# Mechanical stretching of pulmonary vein stimulates matrix metalloproteinase-9 and transforming growth factor-β1 through stretch-activated channel/MAPK pathways in pulmonary hypertension due to left heart disease model rats

Hui Zhang[1]*, Wenhui Huang[1], Hongjin Liu[1], Yihan Zheng[1], Lianming Liao[2]

1 Department of Cardiac Surgery, Union Hospital, Fujian Medical University, Fuzhou, Fujian Province, P.R. China, 2 Department of Medical Laboratory, Union Hospital, Fujian Medical University, Fuzhou, Fujian Province, P.R. China

* zhanghui9877656@163.com

## Abstract

Pulmonary hypertension due to left heart disease (PH-LHD) is a momentous pulmonary hypertension disease, and left heart disease is the most familiar cause. Mechanical stretching may be a crucial cause of vascular remodeling. While, the underlining mechanism of mechanical stretching-induced in remodeling of pulmonary vein in the early stage of PH-LHD has not been completely elucidated. In our study, the PH-LHD model rats were successfully constructed. After 25 days, doppler echocardiography and hemodynamic examination were performed. In addition, after treatment, the levels of matrix metalloproteinase-9 (MMP-9) and transforming growth factor-β1 (TGF-β1) were determined by ELISA, immunohistochemistry and western blot assays in the pulmonary veins. Moreover, the pathological change of pulmonary tissues was evaluated by H&E staining. Our results uncovered that left ventricular insufficiency and interventricular septal shift could be observed in PH-LHD model rats, and the right ventricular systolic pressure (RVSP) and mean left atrial pressure (mLAP) were also elevated in PH-LHD model rats. Meanwhile, we found that MMP-9 and TGF-β1 could be highly expressed in PH-LHD model rats. Besides, we revealed that stretch-activated channel (SAC)/mitogen-activated protein kinases (MAPKs) signaling pathway could be involved in the upregulations of MMP-9 and TGF-β1 mediated by mechanical stretching in pulmonary vein. Therefore, current research revealed that mechanical stretching induced the increasing expressions of MMP-9 and TGF-β1 in pulmonary vein, which could be mediated by activation of SAC/MAPKs signaling pathway in the early stage of PH-LHD.

**Data Availability Statement:** All relevant data are within the manuscript and its Supporting Information files.

**Funding:** This work was supported by the National Natural Science Foundation of China [grant numbers 81770368]; and Medical Elite Cultivation Program of Fujian, P.R.C [grant number 2015-ZQNZD- 16]. The funders participated in study design, data collection, decision to publish, and preparation of the manuscript.

**Competing interests:** No authors have competing interests.

**Abbreviations:** MAPK, Mitogen-activated protein kinase; MMP-9, matrix metalloproteinase-9; PH, pulmonary hypertension; PH-LHD, pulmonary hypertension due to left heart disease; RHC, right heart catheter; RVSP, right ventricular systolic pressure; SAC, stretch-activated channel; TGF-β1, transforming growth factor-β1.

## Introduction

Pulmonary hypertension due to left heart disease (PH-LHD) belongs to type II pulmonary hypertension (PH) and is the most common type of PH in clinical practice [1, 2]. It is usually initiated by left ventricular dysfunction, which results in increased left ventricular end-diastolic pressure, increased left atrial pressure, and finally increased pulmonary venous pressure. This eventually leads to PH and pulmonary artery revascularization [1, 3]. Thus pulmonary vascular remodeling is the pathological basis of PH-LHD [4]. Study has demonstrated that the earliest pathophysiological change occurs in the pulmonary veins during the disease process of PH-LHD [5]. Whether pulmonary venous hypertension in the early stage of the disease participates in pulmonary artery remodeling has not been reported.

Cell adhesion, spreading, growth and differentiation can be regulated by various physical and chemical factors, among which mechanical factors have a great influence on these biological behaviors of cells [6, 7]. Previous studies have indicated that the stretch of cell membrane and its surface receptors can cause the changes of cell morphology and nuclear position, thus affecting cell growth [8, 9]. Research has also testified that cell morphology and spreading degree can affect gene expression and even determine cell survival and death [10, 11]. Currently, Mechanical stretching has become a crucial factor in regulating the structure and function of mammalian cells and tissues, but excessive mechanical stretching will cause vascular remodeling [12]. Nevertheless, the functions and mechanisms of mechanical stretching on PH-LHD has not been entirely elucidated.

Matrix metalloproteinase-9 (MMP-9), also known as gelatinase-B, is one of the most crucial members of the MMPs family [13]. Long-term hypertension can lead to vascular endothelial damage and stimulate smooth muscle to produce MMP-9 [14]. MMP-9 can also degrade the vascular extracellular matrix (ECM), damage the vessel wall and lead to vascular remodeling [15]. Transforming growth factor-β1 (TGF-β1), as a pleiotropic factor, is the most dominant subtype of TGF-β [16, 17]. TGF-β1 has been proven to protect the aorta and other blood vessels from damage caused by factors such as high blood pressure and high cholesterol [18, 19]. Researches have shown that MMP-9 and TGF-β1 play vital roles in pulmonary vascular remodeling [14, 20, 21]. Study showed that the enhanced TGF-β-driven smad-2/3 signaling promoted pulmonary vascular remodeling in inflaming lungs [22]. Study proved that the increased activity and expression of MMP-9 in Pulmonary arterial hypertension (PAH), and effective blocking of MMP-9 could provide an option in the therapeutic intervention of human PAH [23].

SAC, serves a mechanically-sensitive ion channel, can be regulated by the changes in the tension of a cellular scaffold or lipid membrane coupled with channel proteins [24]. It was reported that SAC has significant effect in the electrophysiological changes of the myocardium during stimulation [25]. MAPK pathway is proven to be involved in various physiological processes, such as cell growth, development and inter-cell function synchronization, which also can be closely related to the regulation of inflammation and stress response [26, 27]. Studies also testified that mechanical stretching can activate the phosphorylation of the MAPK pathway [28, 29]. Also, studies have shown that the expression of MMP-9 and TGF-β1 in tissues may be related to stretch-activated channel (SAC) and mitogen-activated protein kinase (MAPK) signaling pathway [12, 30, 31]. Therefore, we speculated that SAC and MAPK signaling pathway may be potential regulatory molecules in PH-LHD.

In this study, the isolated pulmonary veins of PH-LHD rats were mechanically stretched and the expression of MMP-9 and TGF-β1 that corresponded to the early stage of the disease was evaluated. Also, the roles of SAC and MAPK signaling pathway on the expression of MMP-9 and TGF-β1 were explored to investigate the possible mechanism of pulmonary

vascular remodeling. The study may shed new light on the prevention and treatment of PH-LHD.

## Materials and methods

### Animals

Male Sprague Dawley (SD) rats (3–4 weeks old and weighing 80–100 g; license number SCXK (Hu) 2017–0005) were provided by the Shanghai Slack Laboratory Animal Co. (Shanghai, China). Before the experiment, all rats were kept adaptively for one week, fed a standard diet and fed free water. All rats in this study have been approved by the ethics committee of Fujian Medical University. All animal experiments were in compliance with the Guide for the Care and Use of Laboratory Animals published by the US National Institutes of Health and the Animal Care and Use committees of Fujian Medical University (NO. FJMU IACUC 2018–067).

### Establishment of PH-LHD model

PH-LHD model was established by banding procedure described by Siegfried Breitling [32]. Banding is a coarctation procedure used for the ascending aorta above the coronary artery. It is an ideal method to establish model for congestive heart failure and pulmonary arterial hypertension caused by left heart disease [32–34]. It is widely used in experimental research of PH [35–38]. The rats were intraperitoneally injected with 2% sodium pentobarbital (0.3 ml/ 100 g) and subject to non-invasive nasal mask ventilation. The thorax was opened at the second intercostal space on the left side of the sternum to expose the ascending aorta. The metal titanium clip (Hemoclip®; Weck Closure System, Research Triangle Park, NC) with a preset internal diameter of 0.8 mm was used to clamp the ascending aorta. During the first three days after surgery, subcutaneous injection of Carprofen (4 mg/kg bw; Rimadyl TM, Pfizer Gmb H, Karslruhe, Germany) was given daily for postoperative analgesia. The rats in the sham group also underwent operation to expose the ascending aorta. But the ascending aorta was not clamped. The remaining postoperative treatment was the same as in the model group. When the rats reached the humane end, they were euthanized through decapitation.

### Mechanical strength

The isolated vascular tone tester 620M bath (Danish Myo Technology A/S, Denmark) was preheated to 37˚C with 8 ml of K-H solution. Pulmonary vein ring was hanged on the stainless steel hook and adjusted to zero calibration. The K-H solution was changed every 15 minutes during the experiment and a continuous mixture of 95% $O_2$ and 5% $CO_2$ were provided. the vessel rings were first kept in K-H solution for 60 minutes. And then a mechanical strength of 2.0 g was applied for 60 minutes, while in the S1 and M1 groups no mechanical strength was applied. Finally, the above treated pulmonary vein rings were collected and stored in a -80˚C freezer.

### Group of experiments

A total of 48 SD rats were randomly divided into the sham group (n = 12) and the PH-LHD group (n = 36). Corresponding to different inhibitors and mechanical strength used to treat pulmonary venous rings, the sham group was further divided into two subgroups: Sham 1 (S1, mechanical strength: 0 g) and Sham 2 (S2, mechanical strength: 2.0 g); and the PH-LHD group was also further divided into 2 subgroups: Model 1 (M1, banding+0 g), Model 2 (M2, banding +2.0 g). In addition, a broad spectrum inhibitor of MEK1 and MEK2 (U0126, HY-12031 Med-Chem Express, USA), p38 MAPK inhibitor (SB203580, HY-10256 MedChem Express, USA),

SAC inhibitor (Streptomycin, HY-B0472 MedChem Express, USA) and broad-spectrum inhibitor of JNK1, JNK2, and JNK3 (SP600125, HY-12041 MedChem Express, USA) were dissolved in dimethyl sulfoxide (DMSO, HY-10999 MedChem Express, USA), and diluted with K-H solution to 250 μmol/L, 200 μmol/L, 1000 μmol/L, and 250 μmol/L, respectively [12]. Specifically, the pulmonary veins were immersed in the K-H solution containing the corresponding inhibitors (SB203580, SP600125, U0126 or Streptomycin) at 37˚C under the condition of continuous oxygen infusion for 60 mins, and then was stretched with 2.0 g mechanical strength for 60 mins.

## Doppler echocardiography

On day 25 after surgery, Doppler echocardiography was performed to confirm that the metal titanium clip did not fall off or displaced. The diameter of the pulmonary artery and the right ventricular end-diastolic diameter were measured and the left cardiac function was evaluated. Right ventricular systolic pressure (RVSP) was monitored with a right heart catheter (RHC), mean left atrial pressure (mLAP) was monitored directly, and data was collected by BL-420S experimental system of biological function (Chengdu Taimen Software Co., Ltd). PH-LHD was confirmed by left heart failure and increased RVSP.

## Sample preparation

On day 25, all rats were anesthetized with 2% pentobarbital sodium (ip., 0.3 ml/100 g) after Doppler echocardiography and hemodynamic examination. The blood of inferior vena cava was collected and centrifuged (2000 ×g, at 4˚C for 5 mins), and then the collected serum was stored in the refrigerator at -80˚C. The chest was opened in the middle to confirm that the metal titanium clip was on the ascending aorta. The prepared PBS at 4˚C was slowly injected from the right ventricle, and then to the left atrium through the pulmonary circulation; and then the remaining blood in the rat pulmonary circulation was discharged until the pulmonary lobe became white. The heart was quickly separated, dried and weighed. Besides, the pulmonary vein, pulmonary artery, and lung tissue were removed, and the pulmonary vein was placed in a Krebs-Henseleit solution (118 mmol/L NaCl, 4.70 mmol/L KCl, 1.20 mmol/L $MgSO_4$, 25.00 mmol/L NaHCO3, 1.20 mmol/L $KH_2PO_4$, 11.0 mmol/L glucose, 0.50 mmol/L EDTA-$Na_2$, and 2.50 mmol/L $CaCl_2$) at 4˚C. The peripheral tissues of the pulmonary veins were removed under a microscope and 4-mm pulmonary venous rings were prepared (S1 Fig). Pulmonary artery, pulmonary vein and peripheral lung tissues were collected and stored in a -80˚C freezer. Moreover, the pulmonary lobes were removed and immobilized in 4% paraformaldehyde for 24 h and transferred to 75% alcohol.

## Western blotting analysis

For measuring the levels of MMP-9 and TGF-β1, tissues were fully ground into homogenate with RIPA buffer and protease inhibitor (Beyotime Institute of Biotechnology, China) and centrifuged for 20 min at 12000 rpm at 4˚C. The supernatant was collected and the protein concentration was measured using a BCA protein assay Kit (Beijing Solarbio Science & Technology Co., Ltd. China). Protein samples (40 μg for each) were separated by SDS-PAGE and then transferred onto the PVDF membranes. After being blocked in 5% BSA in TBS buffer for 2.0 hrs, the membranes were incubated with the primary antibodies (MMP-9 or TGF-β1, 1:1000, rabbit polyclonal antibody, Abcam) overnight at 4˚C. After washing 3 times with 1× TBST, the membranes were incubated with goat anti-rabbit IgG (1:5000, Affinity) for 1.0 hours at room temperature. The specific bands were developed after washing 3 times by

1 × TBST. Finally, the bands were visualized by ChemiDoc Imaging Systems, and the intensity of protein bands were quantified using Image J software (NIH, Bethesda, MD, USA).

## ELISA assay

Immediately after removal on day 25, venous blood samples were centrifuged for 20 min (1200 revolutions/ min [rpm], 4˚C), and serum was stored at –20˚C until analysis. Serum MMP-9 and TGF-β1 levels were determined with a commercial immunoassay kit specifc to rat MMP-9 and TGF-β1 (Neobioscience, Inc., CN) as per the manufacturer's instructions. This kit is a highly sensitive two-site enzyme-linked immunosorbent assay (ELISA) for measuring MMP-9 and TGF-β1. All experiments were performed in triplicate.

## Immunohistochemistry (IHC) assay

The lung tissues were dipped into 10% Formalin solution for 24 h. The monoclonal antibody against MMP-9 or TGF-β1 (1:200, Abcam, Cambridge, MA, USA) and the goat anti-rabbit-IgG (Abcam, Cambridge, MA, USA) were used for immunostaining. Negative and positive controls were used for both staining procedures.

## Hematoxylin-eosin (H&E) staining

The lung tissues were dehydrated in ethanol solution, and transparentized in dimethylbenzene. After paraffin embedding, tissues were cut into slices. Then slices were roasted, dewaxed and hydrated, incubated in hematoxylin solution, cultured in alcohol and hydrochloric acid and cultured in Scott blue buffer. Finally, they were dehydrated, transparentized, mounted, and observed under a microscope (CKX41, Olympus, Japan). Each tissue of model group and sham group photographed with 200-fold and 400-fold field of view.

## Statistical analysis

The data were expressed as mean ± Standard Error of Mean (mean±SEM), and the unpaired t-student tests or one-way ANOVA were used to compare the data in our study. $P < 0.05$ was considered to be statistically significant. The statistical analysis and mapping were performed using the SPSS17.0 and Graph Pad Prism 5.0.

## Results

### Establishment of PH-LHD model

According to previous reports [32–34], PH-LHD model was established using the banding. Firstly, we exhibited the position of the titanium clip in the ascending aorta above the coronary artery during surgery (Fig 1A). After 25 days, the position of metal titanium clip was evaluated using the Doppler echocardiography, and the graphical result displayed that we have successfully established the PH-LHD model rats by putting the metal titanium clip in the right place (Fig 1B).

### Doppler echocardiography of left cardiac function in PH-LHD model rats

In addition, to further determine the changes of left cardiac function in PH-LHD model rats, doppler echocardiography was adopted to assess the left ventricular dysfunction and the change of interventricular septum. As presented in Fig 2 and Table 1, the diastolic ventricular septum (IVSd), diastolic left ventricular posterior wall (LVPWd), diastolic left ventricular inner diameter (LVDd) and systolic left ventricular inner diameter (LVDs) in the PH-LHD

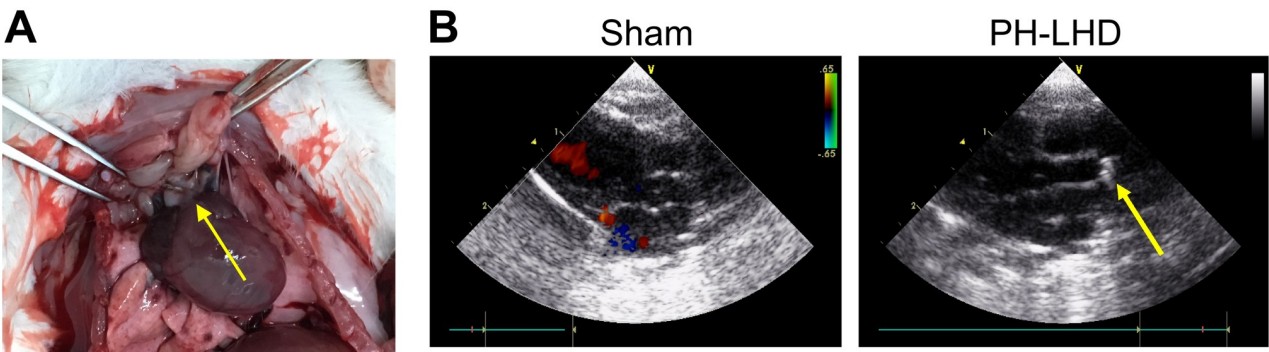

**Fig 1. Establishment of PH-LHD model.** (A) PH-LHD model was constructed, and the position of the titanium clip was displayed using the yellow arrow in the ascending aorta above the coronary artery. (B) The metal titanium clip (yellow arrow) was presented at the ascending aorta of the PH-LHD group by doppler echocardiography.

group were greater than those in the sham group ($P<0.05$). The ejection fraction (EF) in the PH-LHD group was lower than that in the sham group ($P<0.05$). There were no significant differences in heart rate (HR), ascending aorta diameter (AAO), diastolic left atrial inner diameter (LADd), diastolic right ventricular inner diameter (RVDd), and pulmonary artery inner diameter (Pad) between the PH-LHD group and the sham group ($P>0.05$).

### RVSP and mLAP were determined in PH-LHD model rats

Increased pulmonary arterial pressure is the gold standard for the diagnosis of PH. Right heart failure occurs as a late complication of PH. RHC was used to measure RVSP and evaluate the degree of PH [39]. On day 25 after banding, RVSP and mean left atrial pressure (mLAP) in the PH-LHD group was higher than those in the sham group ($P<0.05$). In addition, compared with the sham group, rats in the PH-LHD group demonstrated a significant increase in the heart weight and heart weight/body weight ratio ($P<0.05$), and a significant decrease in the body weight ($P<0.05$). And They are similar in the Right ventricular/(left ventricular+ interventricular septum) ($P<0.05$, Fig 3 and Table 2).

### The pathological changes of pulmonary tissues in PH-LHD model rats

Subsequently, the pathologic structures of pulmonary tissues were identified by H&E staining in PH-LHD model rats after 25 days. Our results exhibited that in the sham group, there were smooth wall of small pulmonary artery, thin pulmonary artery walls, large lumen, evenly distributed cells, and no inflammatory infiltration around the blood; in the PH-LHD group, we discovered that the walls of the small pulmonary artery were slightly thickened, the cells were orderly, and a small amount of inflammatory cells were observed around the vessels (Fig 4).

### High expressions of MMP-9 and TGF-β1 in pulmonary veins

On the mechanism, we then investigated the MMP-9 and TGF-β1 expressions in pulmonary artery, pulmonary vein and pulmonary tissue. And the results of western blotting analysis uncovered that the levels of MMP-9 and TGF-β1 were prominently elevated in pulmonary vein relative to pulmonary artery or pulmonary tissue ($P<0.01$, $P<0.001$, Fig 5).

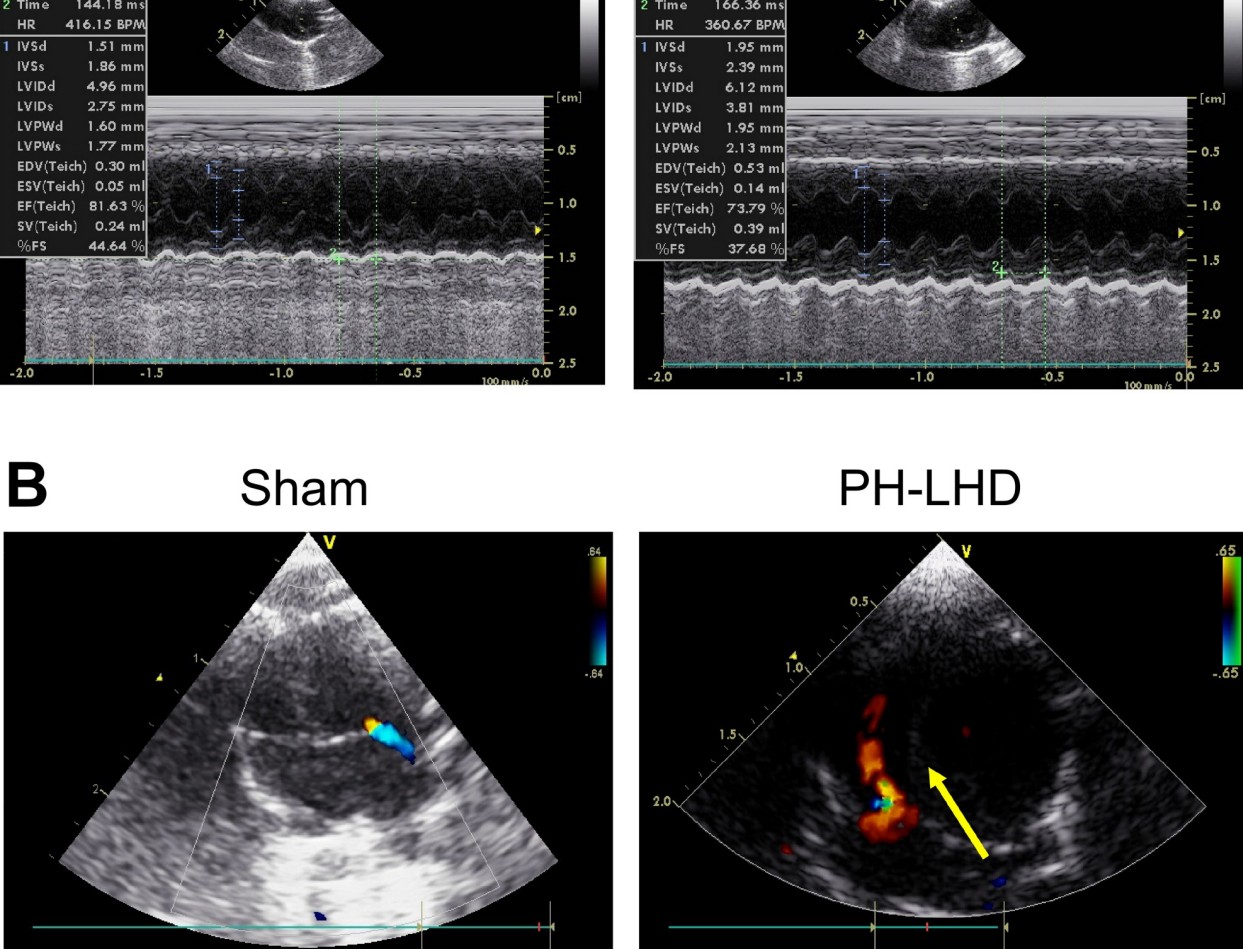

**Fig 2. Doppler echocardiography of left cardiac function in PH-LHD model rats.** (A) Representative images of left ventricular dysfunction in rats were assessed by doppler echocardiography in sham and PH-LHD model rats. (B) The change of interventricular septum in the PH-LHD group (yellow arrow) was evaluated using the doppler echocardiography.

## MMP-9 and TGF-β1 were highly expressed in PH-LHD model rats

Our previous experiments prove that MMP-9 and TGF-β1 were highly expressed in pulmonary veins, and we further explored the expressions of MMP-9 and TGF-β1 in mechanical stretching-induced remodeling of pulmonary veins in the early stage of PH-LHD. Firstly, we certified that the venous blood of rats was collected for ELIAS assay, and the results showed that the expressions of MMP-9 and TGF-β1 were significantly higher in the PH-LHD group than those in the sham group on day 25 after banding ($P<0.01$, $P<0.001$, Fig 6A). Secondly, immunohistochemical results of pulmonary tissues also showed that the expression levels of MMP-9 and TGF-β1 were significantly increased in the pulmonary tissues of the PH-LHD group with respect to that in the sham group. Besides, in the sham group, a few alveolar epithelial cells were positively expressed, with a low degree of staining; in the PH—LHD group, large Numbers of MMP-9 and TGF-β1 positive cells were observed around the pulmonary vein (Fig 6B). Finally, we disclosed that MMP-9 and TGF-β1 expressions were remarkably increased in

**Table 1. Differences in IVSd, LVPWd, LVDd, LVDs, EF, HR, AAO, LADd, RVDd, Pad between sham-operated and PH-LHD model groups.**

| Groups | Sham control group (N = 12) | PH-LHD group (N = 36) | P value |
|---|---|---|---|
| IVSd (mm) | 1.518±0.021 | 1.924±0.009 | <0.001* |
| LVPWd (mm) | 1.618±0.016 | 1.957±0.017 | <0.001* |
| LVDd (mm) | 4.873±0.096 | 5.288±0.093 | 0.019* |
| LVDs (mm) | 2.927±0.079 | 3.222±0.056 | 0.008* |
| EF (%) | 78.31±0.537 | 74.93±0.287 | <0.001* |
| HR (bpm) | 376.7±10.94 | 390.8±6.80 | 0.297 |
| AAO (mm) | 2.133±0.014 | 2.189±0.018 | 0.103 |
| LADd (mm) | 3.367±0.022 | 3.294±0.026 | 0.129 |
| RVDd (mm) | 2.117±0.021 | 2.153±0.019 | 0.317 |
| Pad (mm) | 2.233±0.037 | 2.222±0.021 | 0.792 |

* $P<0.05$, diastolic ventricular septum (IVSd), diastolic left ventricular posterior wall (LVPWd), diastolic left ventricular inner diameter (LVDd), systolic left ventricular inner diameter (LVDs), ejection fraction (EF), heart rate (HR), ascending aorta diameter (AAO), diastolic left atrial inner diameter (LADd), diastolic right ventricular inner diameter (RVDd), pulmonary artery inner diameter (Pad).

the pulmonary vein tissues of the PH-LHD group with versus that in the sham group, and 2.0 g intension was more significant for the MMP-9 and TGF-β1 expressions than the 0 g intension ($P<0.05$, Fig 6C and 6D).

## Mechanical stretching upregulated MMP-9 and TGF-β1 in pulmonary vein by SAC/MAPKs signaling pathway

To further inquiry the underlining mechanism of mechanical stretching on the upregulations of MMP-9 and TGF-β1 in pulmonary vein, we adopted MAPK pathway inhibitors (SB203580, SP600125, U0126) and stretch-activated channel (SAC) inhibitor (streptomycin). The results of western blotting analysis uncovered that compared with the control group, the levels of MMP-9 and TGF-β1 were notably reduced in PH-LHD model rats after treatment with SB203580, SP600125, U0126 and Streptomycin ($P<0.01$, $P<0.001$, Fig 7). The results suggested that mechanical stretching of pulmonary vein increased the expressions of MMP-9 and TGF-β1, while the inhibitors of SAC/MAPKs signaling pathway (SB203580, SP600125, U0126 and streptomycin) could reverse the increases of MMP-9 and TGF-β1 expressions. Therefore, we revealed that SAC and MAPKs signaling pathways might be participate in the promoting effects of mechanical stretching of pulmonary vein on the pulmonary hypertension.

## Discussion

PH-LHD is the most common type of PH. Due to its unclear pathogenesis and complicated pathophysiological process [5], there is still no effective treatment available. Pulmonary vein hypertension is the earliest pathophysiological change in the development of PH-LHD. It is hypothesized that early changes of pulmonary vein wall will promote subsequent pulmonary vascular remodeling in PH-LHD. If this hypothesis is confirmed, it will provide novel targets for the early prevention and treatment of PH-LHD.

Previously Breitling et al found that distinct vascular remodeling occurred in the pulmonary artery within two months after banding [38]. In the present study, Doppler echocardiography on day 25 after banding confirmed the presence of left heart failure in rats, as indicated

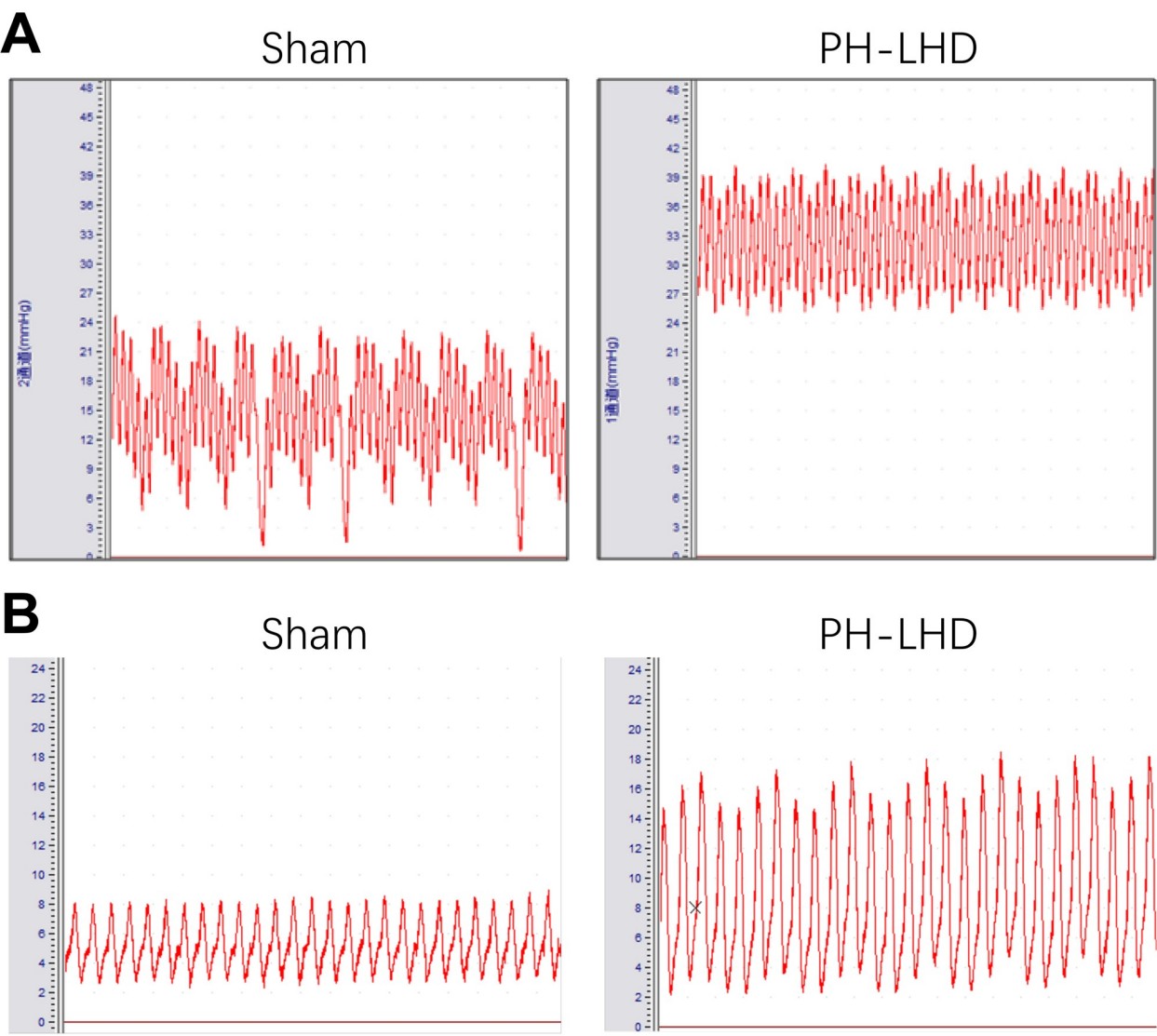

**Fig 3. RVSP and mLAP were determined in PH-LHD model rats.** The representative images of RVSP (A) and mLAP (B) in rats were demonstrated in the sham and PH-LHD model rats.

by increased left ventricular pressure, increased left ventricular diameter, interventricular septum shift, and decreased ejection fraction. Meanwhile, RVSP and mLAP were increased. Thus, although pulmonary blood vessels have not been remodeled at this time point, pulmonary artery pressure has increased. Therefore, we collected pulmonary veins on day 25 after banding, which corresponds to the early stages of PH-LHD, and explore their effects on pulmonary vascular remodeling and possible mechanisms.

Studies have demonstrated that MMP-9 and TGF-β1 are also important molecules in the pathogenesis of PH [23, 40]. The former is involved in the fibrosis and remodeling of pulmonary blood vessels, while the latter is involved in pulmonary vascular remodeling and pulmonary hypertension by regulating smooth muscle cell proliferation and fibroblast phenotypic conversion [23, 41]. Therefore, in hypoxia-induced PH, the expression of MMP-9 and TGF-β1 in the pulmonary arteries increases [41, 42]. In PH-LHD, during the early stages of PH, the pulmonary arteries have not yet undergone vascular remodeling. Increased mechanical

**Table 2. Differences of RVSP, mLAP, BW, HW, HW/BW, RV/(LV+S) between sham groups and PH-LHD model groups.**

| Groups | Sham control group (N = 12) | PH-LHD group (N = 36) | *P* value |
|---|---|---|---|
| RVSP (mmHg) | 24.43±0.51 | 35.77±0.45 | <0.001* |
| mLAP (mmHg) | 4.88±0.42 | 8.01±0.44 | <0.001* |
| BW (g) | 254.2±1.19 | 218.9±6.47 | <0.001* |
| HW (g) | 0.329±0.011 | 0.579±0.008 | <0.001* |
| HW/BW (%) | 0.131±0.004 | 0.257±0.002 | <0.001* |
| RV/(LV+S) | 0.1075±0.006 | 0.1031±0.010 | 0.734 |

Right ventricular systolic pressure (RVSP), Mean left atrial pressure (mLAP), Body weight (BW), Heart weight (HW), Right ventricular (RV), left ventricular (LV), interventricular septum (S).

tension in the pulmonary vein wall is caused by blockage of blood flow in the pulmonary vein in the left heart disease, i.e., pulmonary venous hypertension. Is there any pathological change in pulmonary vein tissue at this time that contributes to the later pulmonary artery vascular remodeling? We found that on day 25 after banding, the expression levels of MMP-9 and TGF-β1 in the pulmonary vein tissue increased compared to the sham group. Shear stress, expansion force, and blood pressure of the pulmonary vein wall may lead to cytoskeletal reorganization and morphological changes. Correspondingly, vascular smooth muscle cells

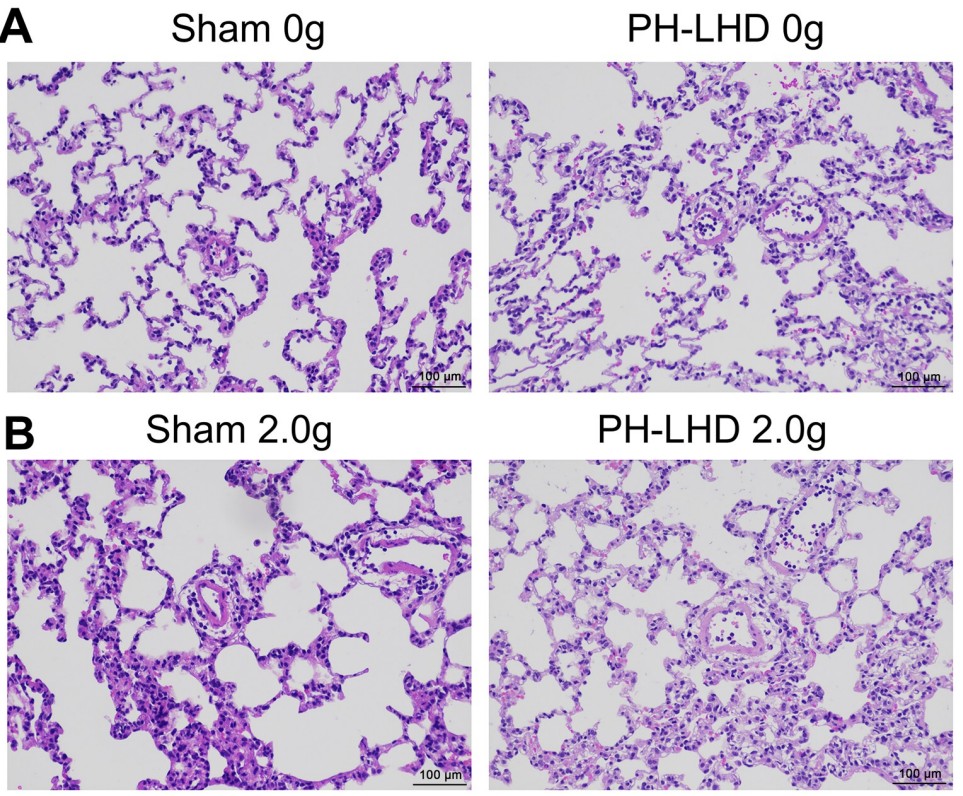

**Fig 4. The pathological changes of pulmonary tissues in PH-LHD model rats.** (A–B) The pathologic structures were confirmed by H&E staining in the sham and PH-LHD model rats treated and untreated 2.0 g. Magnification, 200×; Scale bar = 100 μm. Each group contained six rats.

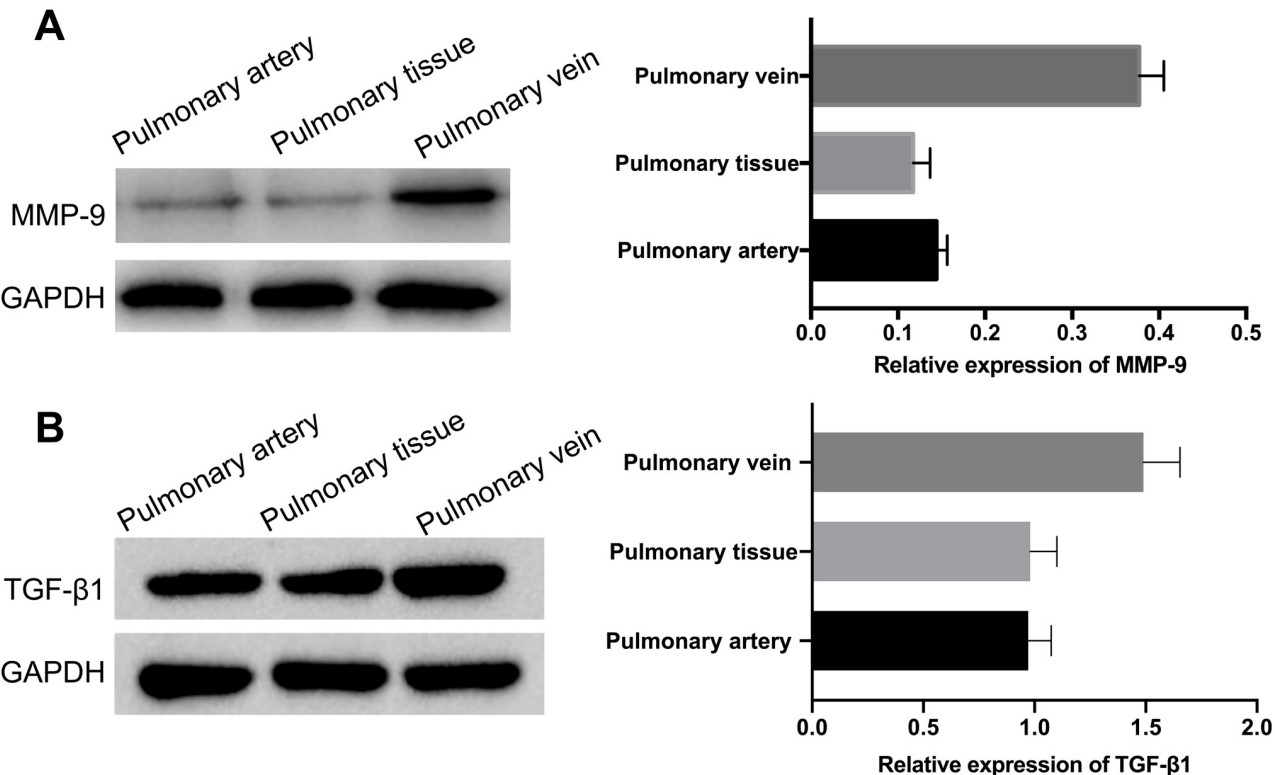

**Fig 5. High expressions of MMP-9 and TGF-β1 in pulmonary veins.** Western blotting analysis was performed to examine the levels of MMP-9 and TGF-β1 in pulmonary artery, pulmonary vein and pulmonary tissue, respectively. GAPDH acts as an internal reference. $^{**}P<0.01$, $^{***}P<0.001$ vs. pulmonary artery group. Each group contained six rats.

produced various molecules, including MMP-9 and TGF-β1, that acted on themselves to maintain and adapt to the pathological changes of vascular structure and function [43].

Recently, Qiu et al demonstrated that mechanical stretching could up-regulate the expression of MMP-9 in the smooth muscle cells from animals with aortic dissection through SAC/NF-κB signaling pathway [30]. The mechanical stress of the vascular wall during aortic dissection may influence the structure and function of cells, tissues and organs [44]. Once this is sensed by cells, it is then transmitted to the nucleus through intracellular signaling pathways, thereby causing the expression of related genes and leading to different physiological or pathological responses [45]. So, does this mechanism of stretch also occur during the increased mechanical tension in the pulmonary vein wall in PH-LHD? In order to explore this assumption, we applied mechanical stretching to pulmonary vein rings of PH-LHD rats. The results showed that there was no significant difference in the expression of MMP-9 and TGF-β1 in the pulmonary veins from rats of the sham group even a force of 2.0 g was applied. However, the expression of MMP-9 and TGF-β1 in the pulmonary veins of PH-LHD rats was up-regulated when a force of 2.0 g was applied. These results indicate that during the development of PH-LHD, as the pulmonary vein wall was continuously stimulated by increased mechanical tension, the cells became sensitive to the mechanical tension and the expression of MMP-9 and TGF-β1 in the pulmonary vein rings were further increased once they were subject to the mechanical stress *in vitro*.

Mechanical stretching has been shown to activate multiple signaling pathways that are related to some pathophysiological processes. For example, Ghantous and his colleagues

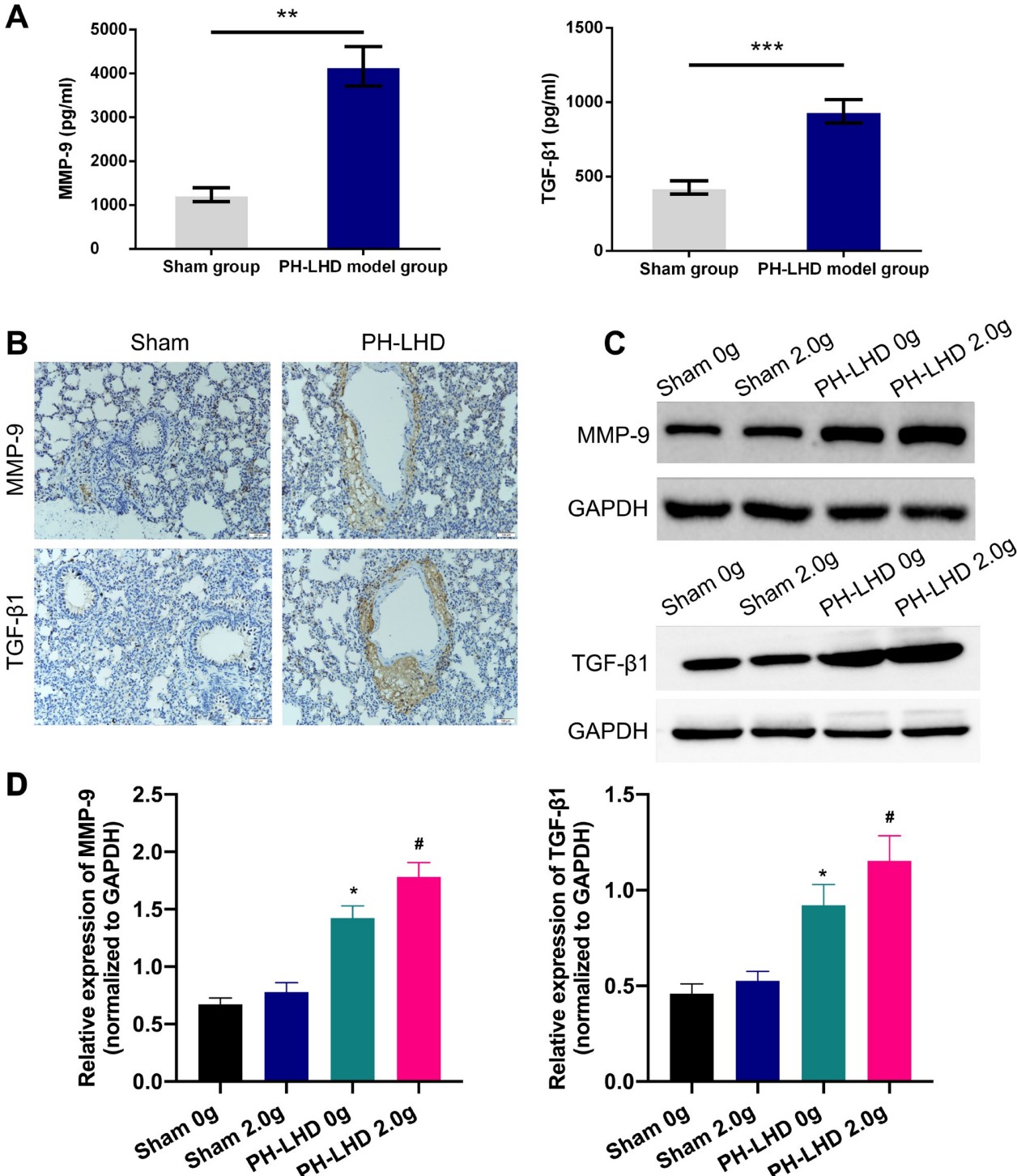

**Fig 6. MMP-9 and TGF-β1 were highly expressed in PH-LHD model rats.** (A) The concentrations of MMP-9 and TGF-β1 were analyzed by ELISA assay in the serums of sham and PH-LHD model rats, $^{**}P<0.01$, $^{***}P<0.001$ vs. sham group. (B) IHC experiment was carried out to examine the levels of MMP-9 and TGF-β1 in the pulmonary tissues of sham and PH-LHD model rats. Magnification, 200×; Scale bar = 100μm. (C) The protein expression levels of MMP-9 and TGF-β1 were assessed by western blot analysis in the pulmonary vein tissues of the sham and PH-LHD model rats. (D) Relative expression levels of MMP-9 and TGF-β1 were counted based on the grey values of the western blotting results. $^{*}P<0.05$ vs. sham 0 g group; #$P<0.05$ vs. sham 2.0 g group. Each group contained six rats.

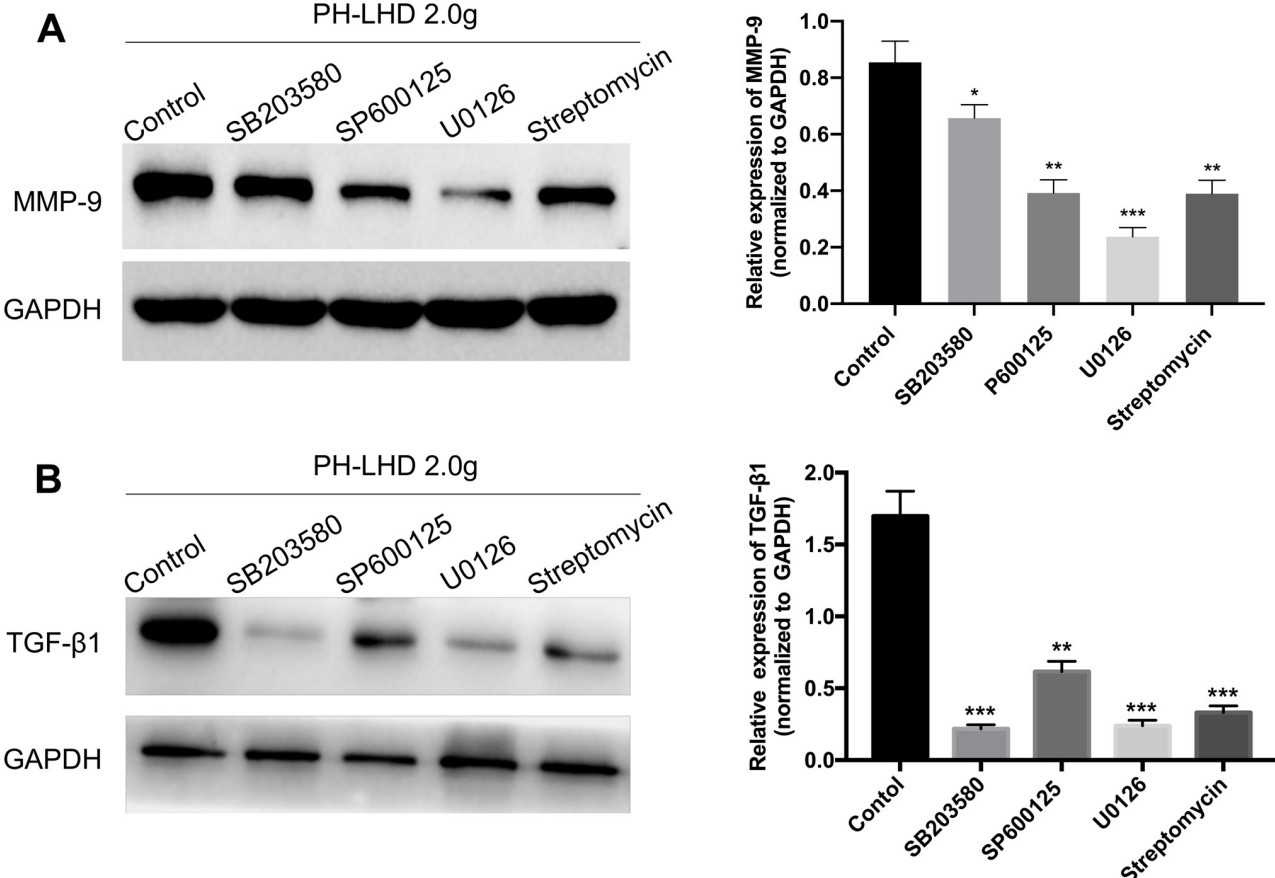

**Fig 7. Mechanical stretching upregulated MMP-9 and TGF-β1 in pulmonary vein by SAC/MAPKs signaling pathway.** PH-LHD model rats were treated with SB203580, SP600125, U0126 and Streptomycin, respectively. (A–B) MMP-9 and TGF-β1 expressions were confirmed by western blot assay, and the relative expressions were calculated based on gray value. $^{**}P<0.01$, $^{***}P<0.001$ vs. pulmonary artery group. Each group contained six rats.

demonstrated that mechanical stretching induced leptin synthesis by activating the ROS signaling pathway, which in turn caused vascular smooth muscle cell remodeling [46]. Seo et al. found that mechanical stretching induced MMP-2 overexpression by activating the vascular smooth muscle Akt signaling pathway and participated in vascular remodeling [47]. Ishise et al. found that mechanical stretching induced fibronectin expression by activating the TRPC3-NF-kappa B axis, resulting in a troublesome wound contracture [48]. The SAC/MAPKs signaling pathway is thought to regulate the development of a variety of cardiovascular diseases [49, 50]. SAC was first discovered in the skeletal muscle cells by Guharay and Sachs. It is an ion channel that is widely found in various cells [51]. MAPKs are a class of serine/threonine protein kinases widely found in mammalian cells, including extracellular signal-regulated kinase (ERK1/2), p38 MAPK, and c-Jun NH2-terminal kinase (JNK). Shaw et al. demonstrated that mechanical stretching activated SAC on vascular smooth muscles, causing a series of responses that led to vascular smooth muscle migration and apoptosis [44]. Liu et al. demonstrated that mechanical stretching mediated vascular remodeling by activating the MAPK signaling pathway in human aortic smooth muscle cells [52]. In the present study, we showed that 1) inhibition of p38 MAPK or JNK signaling pathway can significantly reverse the increased expression of MMP-9 in mechanical stretching-induced PH-LHD pulmonary vein; and 2) inhibition of SAC or JNK signaling pathway can significantly reverse the increased

expression of TGF-β1 in mechanical stretching-induced PH-LHD pulmonary vein. Thus increased mechanical tension of the pulmonary vein wall in PH-LHD may activate the SAC/ MAPKs signaling pathway, which regulate the expression of genes involved in PH-LHD, leading to cell proliferation, differentiation, or apoptosis [53–55]. More importantly, functional changes in the early pulmonary veins of PH-LHD have a subtle influence on subsequent vascular remodeling. If the remodeling of pulmonary arteries during the later stages of the disease can be blocked by early inhibition of MMP-9 and TGF-β1 expression mediated by SAC/ MAPKs signaling pathway deserves further study. Besides, in future research, it is also very significant to further explore the specific regulatory mechanisms of SAC/MAPKs signaling pathway on the MMP-9 and TGF-β1 in pulmonary vein after Mechanical stretching.

## Conclusions

During the early stages of PH-LHD, increase in the mechanical tension of the pulmonary vein wall may cause high expressions of MMP-9 and TGF-β1 in the pulmonary vein tissue through the SAC/MAPKs signaling pathway. Therefore, increased expressions of MMP-9 and TGF-β1 might participate in the subsequent pulmonary artery vascular remodeling. This provides new targets for the early interventions of PH-LHD.

## Supporting information

**S1 Fig. The preparation of pulmonary venous rings.**
(TIF)

**S1 Raw data.**
(ZIP)

**S1 Raw images.**
(PDF)

## Acknowledgments

We thank the Institute of Cardiothoracic Surgery of Fujian Medical University Union Hospital for technical assistance and animal care.

## Author Contributions

**Conceptualization:** Hui Zhang, Hongjin Liu.

**Data curation:** Hui Zhang, Wenhui Huang, Hongjin Liu, Yihan Zheng, Lianming Liao.

**Formal analysis:** Wenhui Huang, Hongjin Liu, Yihan Zheng, Lianming Liao.

**Funding acquisition:** Hui Zhang.

**Investigation:** Hui Zhang.

**Resources:** Hui Zhang, Wenhui Huang, Hongjin Liu, Yihan Zheng.

**Software:** Hui Zhang, Yihan Zheng, Lianming Liao.

**Supervision:** Wenhui Huang.

**Validation:** Hui Zhang, Lianming Liao.

**Visualization:** Hui Zhang, Lianming Liao.

**Writing – original draft:** Hui Zhang.

**Writing – review & editing:** Hui Zhang.

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
