## [Decision Letter · Decision Letter 0]

5 Feb 2020

PONE-D-19-35944

Mechanical stretching of pulmonary vein stimulates matrix metalloproteinase-9 and transforming growth factor-β1 through spindle assembly checkpoint/MAPK pathways in pulmonary hypertension due to left heart disease model rats

PLOS ONE

Dear Dr. Zhang,

Thank you for submitting your manuscript to PLOS ONE. After careful consideration, we feel that it has merit but does not fully meet PLOS ONE’s publication criteria as it currently stands. Therefore, we invite you to submit a revised version of the manuscript that addresses the points raised during the review process.

This study addressed the underlining mechanisms of mechanical stretching-induced remodeling of the pulmonary vein in the early stage of PH-LHD. Current research revealed that mechanical stretching induced the increasing expressions of MMP-9 and TGF-β1 in the pulmonary vein, which could be mediated by activation of the SAC/MAPKs signaling pathway in the early stage of PH-LHD. This issue is interesting and important. Some recommendations and suggestions are as follows:

1. The whole words need to be spelled while abbreviations are used for the first time. Such as RVSP, mLAP, PAH, etc. 

2. RVSP and mLAP were measured in this study, however, the mean pulmonary arterial pressure should be a direct indicator of PAH. Why did not authors check the value? Moreover, in the discussion, the authors wrote: “Thus, although pulmonary blood vessels have not been remodeled at this time point, pulmonary artery pressure has increased”.

3. In figure 4, “in sham group, the cells were evenly distributed and of uniform thickness; in PH-LHD group, the cells presented as a continuum of destruction.”, what kind of “the cells” authors mean for? The typical pathological findings of type Ⅱ PH should be congestive heart failure related.

4. Can the authors describe the method of pul artery, pul vein, and lung tissue harvest in detail?

5. In figure 6C, which part of lung tissue did the sample come from? Was it whole lung tissue or only veins? Were there images of arterioles stained for MMP-9 and TGF-b1 to confirm the data of figure 5?

    6. Make sure the reference of “17” cited in the introduction correct or not.

We would appreciate receiving your revised manuscript by Mar 21 2020 11:59PM. To enhance the reproducibility of your results, we recommend that if applicable you deposit your laboratory protocols in protocols.io, where a protocol can be assigned its own identifier (DOI) such that it can be cited independently in the future. For instructions see: http://journals.plos.org/plosone/s/submission-guidelines#loc-laboratory-protocols

We look forward to receiving your revised manuscript.

Kind regards,

Chuen-Mao Yang

Academic Editor

PLOS ONE

Journal Requirements:

1. Thank you for including your ethics statement:

All rats in this study have been approved by the Fujian Medical University. All animal experiments were in compliance with the National Institutes of Health Guide for Care and Use of Laboratory Animals.

Please amend your current ethics statement to confirm that your named ethics committee specifically approved this study.

For additional information about PLOS ONE submissions requirements for ethics oversight of animal work, please refer to http://journals.plos.org/plosone/s/submission-guidelines#loc-animal-research  

2.

PLOS ONE now requires that authors provide the original uncropped and unadjusted images underlying all blot or gel results reported in a submission’s figures or Supporting Information files. This policy and the journal’s other requirements for blot/gel reporting and figure preparation are described in detail at https://journals.plos.org/plosone/s/figures#loc-blot-and-gel-reporting-requirements and https://journals.plos.org/plosone/s/figures#loc-preparing-figures-from-image-files. When you submit your revised manuscript, please ensure that your figures adhere fully to these guidelines and provide the original underlying images for all blot or gel data reported in your submission. See the following link for instructions on providing the original image data: https://journals.plos.org/plosone/s/figures#loc-original-images-for-blots-and-gels.

3. Thank you for including the following funding information within the acknowledgementss ection of your manuscript to read; " This work was supported by the National Natural Science Foundation of China [grant numbers 81770368]; and Medical Elite Cultivation Program of Fujian, P.R.C [grant number 2015-ZQN-ZD-16]. "

"The funders participated in study design, data collection, decision to publish, and preparation of the manuscript."

4.

We note that you have indicated that data from this study are available upon request. PLOS only allows data to be available upon request if there are legal or ethical restrictions on sharing data publicly. For more information on unacceptable data access restrictions, please see http://journals.plos.org/plosone/s/data-availability#loc-unacceptable-data-access-restrictions.

Reviewers' comments:

Reviewer's Responses to Questions

**Comments to the Author**

1. Is the manuscript technically sound, and do the data support the conclusions?

Reviewer #1: Yes

2. Has the statistical analysis been performed appropriately and rigorously? 

Reviewer #1: Yes

3. Have the authors made all data underlying the findings in their manuscript fully available?

Reviewer #1: Yes

4. Is the manuscript presented in an intelligible fashion and written in standard English?

Reviewer #1: Yes

5. Review Comments to the Author

Reviewer #1: This manuscript addressed that the possible mechanisms of pulmonary hypertension in response to left heart disease. The authors used in vivo rat models to study the participation of SAC/MAPK together with MMP9 and TGF-b1 in pulmonary hypertension due to left heart disease (PH-LHD). The study is advancing but several parts need future addressed and clarified.

1. How many rats in each groups should be provided in figure legends.

2. In figure 4 and 6, it would be much better if the specific pathological changes or specific structures were indicated in H&E staining or IHC data. Moreover, the scale bar should be provided in each panels.

3. In figure 6B, the amplification folds seems not equal in sham group and PH-LHD group. The nuclei stains showed in sham group of figure 6B are much larger than PH-LHD group. Moreover, the background of figure 6B should be showed in equal colors or brightness. And is the data collected here 0 g of mechanical stretching? How do the MMP9 and TGF-b1 IHC stains of study groups with mechanical stretching show?

4. It would be much better if the protein expression of figure 6C are quantified and performs the statistical analysis.

5. In introduction section, the importance of SAC and MAPK pathway should be provided.

6. In method section, where are the inhibitors injected? And the final concentration used are?

7. The expression of SAC or phosphorylation of MAPK pathways should be provided.

8. The manuscript needs further English edition.

6. PLOS authors have the option to publish the peer review history of their article (what does this mean?). If published, this will include your full peer review and any attached files.

Reviewer #1: No

---

## [Author Response · Author response to Decision Letter 0]

19 Jun 2020

Dear reviewers:

We sincerely appreciate your time and critical comments on our manuscript entitled "Mechanical stretching of pulmonary vein stimulates matrix metalloproteinase-9 and transforming growth factor-β1 through stretch-activated channel/MAPK pathways in pulmonary hypertension due to left heart disease model rats" (PONE-D-19-35944). We have revised our manuscript as suggested and answered comments point by point.

Reviewer #1

1. The whole words need to be spelled while abbreviations are used for the first time. Such as RVSP, mLAP, PAH, etc. 

Answer: Thanks very much for your comments. We have spelled the whole words of the abbreviations, which were used for the first time in this study.

2. RVSP and mLAP were measured in this study, however, the mean pulmonary arterial pressure should be a direct indicator of PAH. Why did not authors check the value? Moreover, in the discussion, the authors wrote: “Thus, although pulmonary blood vessels have not been remodeled at this time point, pulmonary artery pressure has increased”.

Answer: Thanks for your constructive suggestion. The ideal condition for this study is to be able to directly monitor pulmonary arterial pressure using a Miller catheter. However, due to the limited conditions, we chose PE catheter to monitor the right ventricular systolic pressure.

The pathophysiological changes of pulmonary arterial hypertension associated with left heart disease are that left heart disease first causes left ventricular dysfunction or failure, then leads to the elevation of pulmonary venous pressure and pulmonary hypertension, and finally results in the right heart failure. Thus, the increase of right ventricular pressure can be caused by the increase of pulmonary artery resistance and pressure, and the increase of pressure pulmonary circulation can be caused by the increase of left ventricular pressure. In our study, PH-LHD model is actually a model of pulmonary hypertension caused by the ascending aorta banding, which can gradually lead to the increased right ventricular pressure and right heart failure. So, the increase in right ventricular pressure can indicate the increase in pulmonary artery pressure at this time, although pulmonary vascular remodeling has not yet occurred.

3. In figure 4, “in sham group, the cells were evenly distributed and of uniform thickness; in PH-LHD group, the cells presented as a continuum of destruction.”, what kind of “the cells” authors mean for? The typical pathological findings of type Ⅱ PH should be congestive heart failure related.

Answer: Thanks very much for your comment. We have re-described the results of Figure 4 as follows: Our results exhibited that in the sham group, there were smooth wall of small pulmonary artery, thin pulmonary artery walls, large lumen, evenly distributed cells, and no inflammatory infiltration around the blood; in the PH-LHD group, we discovered that the walls of the small pulmonary artery were markedly thickened, the intima was rough and incomplete, the cells were arranged in disorder, the number of nuclei was increased, the vasculature was severely myogenous, the lumen area was reduced, and the perivascular inflammatory cell infiltration was present.

4. Can the authors describe the method of pul artery, pul vein, and lung tissue harvest in detail?

Answer: Thanks very much for your comment. We have described in detail the specific collection methods of samples in the revised method, as follows: On day 25, all rats were anesthetized with 2% pentobarbital sodium (ip., 0.3ml/100g) after Doppler echocardiography and hemodynamic examination. The blood of inferior vena cava was collected and centrifuged (2000 ×g, at 4℃ for 5 mins), and then the collected serum was stored in the refrigerator at -80℃. The chest was opened in the middle to confirm that the metal titanium clip was on the ascending aorta. The prepared PBS at 4℃ was slowly injected from the right ventricle, and then to the left atrium through the pulmonary circulation; and then the remaining blood in the rat pulmonary circulation was discharged until the pulmonary lobe became white. The heart was quickly separated, dried and weighed. Besides, the pulmonary vein, pulmonary artery, and lung tissue were removed, and the pulmonary vein was placed in a Krebs-Henseleit solution (118 mmol/L NaCl, 4.70 mmol/L KCl, 1.20 mmol/L MgSO4, 25.00 mmol/L NaHCO3, 1.20 mmol/L KH2PO4, 11.0 mmol/L glucose, 0.50 mmol/L EDTA-Na2, and 2.50 mmol/L CaCl2) at 4°C. The peripheral tissues of the pulmonary veins were removed under a microscope and 4-mm pulmonary venous rings were prepared. Pulmonary artery, pulmonary vein and peripheral lung tissues were collected and stored in a -80°C freezer. Moreover, the pulmonary lobes were removed and immobilized in 4% paraformaldehyde for 24 h and transferred to 75% alcohol.

Figure 1S. The preparation of pulmonary venous rings

5. In figure 6C, which part of lung tissue did the sample come from? Was it whole lung tissue or only veins? Were there images of arterioles stained for MMP-9 and TGF-b1 to confirm the data of figure 5?

Answer: Thanks for your constructive suggestion. We adopted the pulmonary vein tissues in the sham operation groups (Sham 0g and sham 2.0g), and in the model group (PH-LHD 0g and PH-LHD 2.0g). And we have explicated the pulmonary vein tissue in the figure legends of figure 6C. In Figure 5, we uncovered that the levels of MMP-9 and TGF-β1 were prominently elevated in pulmonary vein relative to pulmonary artery or pulmonary tissue. In Figure 6, we proved that the expression levels of MMP-9 and TGF-β1 were significantly increased in the pulmonary tissues of the PH-LHD group with respect to that in the sham group; and MMP-9 and TGF-β1 expressions were remarkably increased in the pulmonary vein tissues of the PH-LHD group with versus that in the sham group, and 2.0g intension was more significant for the MMP-9 and TGF-β1 expressions than the 0 g intension.

6. Make sure the reference of “17” cited in the introduction correct or not.

Answer: Thanks very much for your comment. We redefined and modified the reference 17 in the revised manuscript.

Journal Requirements:

1. Thank you for including your ethics statement:

All rats in this study have been approved by the Fujian Medical University. All animal experiments were in compliance with the National Institutes of Health Guide for Care and Use of Laboratory Animals.

Please amend your current ethics statement to confirm that your named ethics committee specifically approved this study.

For additional information about PLOS ONE submissions requirements for ethics oversight of animal work, please refer to http://journals.plos.org/plosone/s/submission-guidelines#loc-animal-research

Answer: Thanks very much for your comment. We confirmed the ethics statement.

2.

PLOS ONE now requires that authors provide the original uncropped and unadjusted images underlying all blot or gel results reported in a submission’s figures or Supporting Information files. This policy and the journal’s other requirements for blot/gel reporting and figure preparation are described in detail at https://journals.plos.org/plosone/s/figures#loc-blot-and-gel-reporting-requirements and https://journals.plos.org/plosone/s/figures#loc-preparing-figures-from-image-files. When you submit your revised manuscript, please ensure that your figures adhere fully to these guidelines and provide the original underlying images for all blot or gel data reported in your submission. See the following link for instructions on providing the original image data: https://journals.plos.org/plosone/s/figures#loc-original-images-for-blots-and-gels.

Answer: Thanks very much for your comment. We have added the Data Availability statement.

3. Thank you for including the following funding information within the acknowledgementss ection of your manuscript to read; " This work was supported by the National Natural Science Foundation of China [grant numbers 81770368]; and Medical Elite Cultivation Program of Fujian, P.R.C [grant number 2015-ZQN-ZD-16]. "

"The funders participated in study design, data collection, decision to publish, and preparation of the manuscript."

Answer: Thanks very much for your comment. We have modified it.

4.

We note that you have indicated that data from this study are available upon request. PLOS only allows data to be available upon request if there are legal or ethical restrictions on sharing data publicly. For more information on unacceptable data access restrictions, please see http://journals.plos.org/plosone/s/data-availability#loc-unacceptable-data-access-restrictions.

Answer: Thanks very much for your comment. We have added the Data Availability statement.

Answer: Thanks very much for your comment. We have applied the ORCID iD.

Reviewers' comments:

Reviewer's Responses to Questions

Comments to the Author

1. Is the manuscript technically sound, and do the data support the conclusions?

Reviewer #1: Yes

Answer: Thanks very much for your comments.

2. Has the statistical analysis been performed appropriately and rigorously? 

Reviewer #1: Yes

 Answer: Thanks very much for your comments.

3. Have the authors made all data underlying the findings in their manuscript fully available?

Reviewer #1: Yes

 Answer: Thanks very much for your comments.

4. Is the manuscript presented in an intelligible fashion and written in standard English?

Reviewer #1: Yes

 Answer: Thanks very much for your comments.

5. Review Comments to the Author

Reviewer #1: This manuscript addressed that the possible mechanisms of pulmonary hypertension in response to left heart disease. The authors used in vivo rat models to study the participation of SAC/MAPK together with MMP9 and TGF-b1 in pulmonary hypertension due to left heart disease (PH-LHD). The study is advancing but several parts need future addressed and clarified.

1. How many rats in each groups should be provided in figure legends?

Answer: Thanks very much for your comment. In our study, six rats were used in each group. We have added this information in the revised figure legends.

2. In figure 4 and 6, it would be much better if the specific pathological changes or specific structures were indicated in H&E staining or IHC data. Moreover, the scale bar should be provided in each panels.

Answer: Thanks for your constructive suggestion. We have described the pathologic changes in detail in the results section of figure 4 and 6. Moreover, we also added the scale bar in each panel.

3. In figure 6B, the amplification folds seems not equal in sham group and PH-LHD group. The nuclei stains showed in sham group of figure 6B are much larger than PH-LHD group. Moreover, the background of figure 6B should be showed in equal colors or brightness. And is the data collected here 0 g of mechanical stretching? How do the MMP9 and TGF-b1 IHC stains of study groups with mechanical stretching show?

Answer: Thanks for your constructive suggestion. We are very sorry that we put the wrong pictures, and we have also replaced the result graphs in the Sham group of Figure 6B. In figure 6B, IHC experiment was carried out to examine the levels of MMP-9 and TGF-β1 in the pulmonary tissues of sham and PH-LHD model rats (0 g of mechanical stretching). Lung tissue sections contained pulmonary veins, pulmonary arteries, and pulmonary tissues. We first compared the expressions of TGF-β1 and MMP-9 in pulmonary veins, pulmonary arteries, and pulmonary tissues 25 days after Banding surgery. In Figure 6B, the pulmonary tissues were selected in the PH-LHD group, which was unable to be mechanically stretched. The IHC staining results of MMP-9 and TGF-β1 were brown-positive. The experimental results showed that on the 25th day after Banding surgery, MMP-9 and TGF-β1 were first expressed around pulmonary vein.

4. It would be much better if the protein expression of figure 6C are quantified and performs the statistical analysis.

Answer: Thanks very much for your comment. We have quantified and statistically analyzed the protein expression in the revised figure 6C.

5. In introduction section, the importance of SAC and MAPK pathway should be provided.

Answer: Thanks very much for your comment. we have also added the importance of SAC and MAPK pathway in the revised introduction section, as follows: SAC, serves a mechanically-sensitive ion channel, can be regulated by the changes in the tension of a cellular scaffold or lipid membrane coupled with channel proteins[1]. It was reported that SAC has significant effect in the electrophysiological changes of the myocardium during stimulation[2]. MAPK pathway is proven to be involved in various physiological processes, such as cell growth, development and inter-cell function synchronization, which also can be closely related to the regulation of inflammation and stress response[3, 4]. Studies also testified that mechanical stretching can activate the phosphorylation of the MAPK pathway[5, 6]. Also, studies have shown that the expression of MMP-9 and TGF-β1 in tissues may be related to stretch-activated channel (SAC) and mitogen-activated protein kinase (MAPK) signaling pathway[7-9]. Therefore, we speculated that SAC and MAPK signaling pathway may be potential regulatory molecules in PH-LHD.

6. In method section, where are the inhibitors injected? And the final concentration used are?

Answer: Thanks very much for your comment. We have further the processing method as follows: A total of 48 SD rats were randomly divided into the sham group (n=12) and the PH-LHD group (n=36). Corresponding to different inhibitors and mechanical strength used to treat pulmonary venous rings, the sham group was further divided into two subgroups: Sham 1 (S1, mechanical strength: 0 g) and Sham 2 (S2, mechanical strength: 2.0 g); and the PH-LHD group was also further divided into 2 subgroups: Model 1 (M1, banding+0 g), Model 2 (M2, banding+2.0 g). In addition, a broad spectrum inhibitor of MEK1 and MEK2 (U0126, HY-12031 MedChem Express, USA), p38 MAPK inhibitor (SB203580, HY-10256 MedChem Express, USA), SAC inhibitor (Streptomycin, HY-B0472 MedChem Express, USA) and broad-spectrum inhibitor of JNK1, JNK2, and JNK3 (SP600125, HY-12041 MedChem Express, USA) were dissolved in dimethyl sulfoxide (DMSO, HY-10999 MedChem Express, USA), and diluted with K-H solution to 250 μmol/L, 250 μmol/L, 1000 μmol/L, and 250 μmol/L, respectively[8]. Specifically, the pulmonary veins were immersed in the K-H solution containing the corresponding inhibitors (SB203580, SP600125, U0126 or Streptomycin) at 37℃ under the condition of continuous oxygen infusion for 60 mins, and then was stretched with 2.0g mechanical strength for 60mins.

7. The expression of SAC or phosphorylation of MAPK pathways should be provided.

Answer: Thanks for your constructive suggestion. In our study, we have preliminarily proved that mechanical stretching of pulmonary vein increased the expressions of MMP-9 and TGF-β1, while the inhibitors of SAC/MAPKs signaling pathway (SB203580, SP600125, U0126 and streptomycin) could reverse the increases of MMP-9 and TGF-β1 expressions. Therefore, we revealed that SAC and MAPKs signaling pathways might be participate in the promoting effects of mechanical stretching of pulmonary vein on the pulmonary hypertension. We thanks to the reviewer for this excellent question which we should add the expressions of SAC and phosphorylation of MAPK pathways. Because of the sample, time and considerable additional work, we did not add the experiment. In our future research, we will also explore further the specific regulatory mechanisms of SAC/MAPKs signaling pathway on the MMP-9 and TGF-β1 in pulmonary vein after Mechanical stretching. This limitation has also been described in the revised discussion section.

8. The manuscript needs further English edition.

 Answer: Thanks very much for your comment. Our manuscript has been revised with the help of native speaker regarding the deficiencies in English grammar, spelling, and sentence structure.

6. PLOS authors have the option to publish the peer review history of their article (what does this mean?). If published, this will include your full peer review and any attached files.

Do you want your identity to be public for this peer review? For information about this choice, including consent withdrawal, please see our Privacy Policy.

Reviewer #1: No

We appreciate very much for your time in reviewing our manuscript. And the suggestions and comments really helped us a lot. If there is any information I can provide, please don’t hesitate to contact us.

Thank you again for your time and patience. We are looking forward to hearing from you soon.

With kindest regards to you.

Yours sincerely

References

1. Schmidt, C., et al., Stretch-activated two-pore-domain (K(2P)) potassium channels in the heart: Focus on atrial fibrillation and heart failure. Prog Biophys Mol Biol, 2017. 130(Pt B): p. 233-243.

2. Liang, J., et al., Stretch-activated channel Piezo1 is up-regulated in failure heart and cardiomyocyte stimulated by AngII. Am J Transl Res, 2017. 9(6): p. 2945-2955.

3. Guo, Y.J., et al., ERK/MAPK signalling pathway and tumorigenesis. Exp Ther Med, 2020. 19(3): p. 1997-2007.

4. Sun, J. and G. Nan, The Mitogen-Activated Protein Kinase (MAPK) Signaling Pathway as a Discovery Target in Stroke. J Mol Neurosci, 2016. 59(1): p. 90-8.

5. Hu, Y., et al., Mechanical stretch aggravates aortic dissection by regulating MAPK pathway and the expression of MMP-9 and inflammation factors. Biomed Pharmacother, 2018. 108: p. 1294-1302.

6. Fu, S., et al., Effects of Cyclic Mechanical Stretch on the Proliferation of L6 Myoblasts and Its Mechanisms: PI3K/Akt and MAPK Signal Pathways Regulated by IGF-1 Receptor. Int J Mol Sci, 2018. 19(6).

7. Qiu, Z., et al., Mechanical strain induced expression of matrix metalloproteinase-9 via stretch-activated channels in rat abdominal aortic dissection. Medical science monitor: international medical journal of experimental and clinical research, 2017. 23: p. 1268.

8. Hu, Y., et al., Mechanical stretch aggravates aortic dissection by regulating MAPK pathway and the expression of MMP-9 and inflammation factors. Biomedicine & Pharmacotherapy, 2018. 108: p. 1294-1302.

9. Zhang, L., et al., 15‐LO/15‐HETE Mediated Vascular Adventitia Fibrosis via p38 MAPK‐Dependent TGF‐β. Journal of cellular physiology, 2014. 229(2): p. 245-257.

---

## [Editor Report · Decision Letter 1]

24 Jun 2020

Mechanical stretching of pulmonary vein stimulates matrix metalloproteinase-9 and transforming growth factor-β1 through stretch-activated channel/MAPK pathways in pulmonary hypertension due to left heart disease model rats

PONE-D-19-35944R1

Dear Dr. Hui Zhang,

We’re pleased to inform you that your manuscript has been judged scientifically suitable for publication and will be formally accepted for publication once it meets all outstanding technical requirements.

Kind regards,

Chuen-Mao Yang

Academic Editor

PLOS ONE

Additional Editor Comments (optional):

The questions raised by the Reviewers have been carefully corrected and the revised manuscript became acceptable for publication.
---

## [Editor Report · Acceptance letter]

20 Aug 2020

PONE-D-19-35944R1 

Mechanical stretching of pulmonary vein stimulates matrix metalloproteinase-9 and transforming growth factor-β1 through stretch-activated channel/MAPK pathways in pulmonary hypertension due to left heart disease model rats 

Dear Dr. Zhang:

I'm pleased to inform you that your manuscript has been deemed suitable for publication in PLOS ONE. Congratulations! Your manuscript is now with our production department. 

Kind regards, 

on behalf of

Dr. Chuen-Mao Yang 

Academic Editor

PLOS ONE